# Ferroptosis in Hepatocellular Carcinoma: Mechanisms, Drug Targets and Approaches to Clinical Translation

**DOI:** 10.3390/cancers14071826

**Published:** 2022-04-04

**Authors:** Dino Bekric, Matthias Ocker, Christian Mayr, Sebastian Stintzing, Markus Ritter, Tobias Kiesslich, Daniel Neureiter

**Affiliations:** 1Center for Physiology, Pathophysiology and Biophysics, Institute for Physiology and Pathophysiology, Paracelsus Medical University, 5020 Salzburg, Austria; dino.bekric@pmu.ac.at (D.B.); christian.mayr@pmu.ac.at (C.M.); markus.ritter@pmu.ac.at (M.R.); tobias.kiesslich@pmu.ac.at (T.K.); 2Translational Medicine & Clinical Pharmacology, Boehringer Ingelheim Pharma GmbH & Co. KG, 55216 Ingelheim, Germany; matthias.ocker@boehringer-ingelheim.com; 3Medical Department, Division of Hematology, Oncology, and Cancer Immunology, Campus Charité Mitte, Charité University Medicine Berlin, 10117 Berlin, Germany; sebastian.stintzing@charite.de; 4Department of Internal Medicine I, Paracelsus Medical University/University Hospital Salzburg (SALK), 5020 Salzburg, Austria; 5Ludwig Boltzmann Institute for Arthritis and Rehabilitation, Paracelsus Medical University, 5020 Salzburg, Austria; 6Gastein Research Institute, Paracelsus Medical University, 5020 Salzburg, Austria; 7Institute of Pathology, Paracelsus Medical University/University Hospital Salzburg (SALK), 5020 Salzburg, Austria; 8Cancer Cluster Salzburg, 5020 Salzburg, Austria

**Keywords:** ferroptosis, iron, hepatocellular carcinoma, drug development, combinatory treatment

## Abstract

**Simple Summary:**

In recent decades, scientific discoveries brought up several new treatments and improvements for patients suffering from hepatocellular carcinoma (HCC). However, increasing resistance to current therapies, such as sorafenib, worsen the outcome of HCC patients, leading to a search for alternative therapeutic strategies. The term ferroptosis describes a novel form of regulated cell death, which is different from apoptosis and necroptosis in a mechanistical and morphological manner. The main mechanism, which leads to cell death, is lipid peroxidation, caused by iron overload and the accumulation of polyunsaturated fatty acids. Recent studies demonstrate that ferroptosis can hamper the carcinogenesis in several tumor entities and in HCC. Therefore, a better understanding and a deeper insight in the processes of ferroptosis in HCC and the possible application of it in the clinical practice are of extreme importance.

**Abstract:**

Ferroptosis, an iron and reactive oxygen species (ROS)-dependent non-apoptotic type of regulated cell death, is characterized by a massive iron overload and peroxidation of polyunsaturated fatty acids (PUFAs), which finally results in cell death. Recent studies suggest that ferroptosis can influence carcinogenesis negatively and therefore may be used as a novel anti-cancer strategy. Hepatocellular carcinoma (HCC) is a deadly malignancy with poor chances of survival and is the second leading cause of cancer deaths worldwide. Diagnosis at an already late stage and general resistance to current therapies may be responsible for the dismal outcome. As the liver acts as a key factor in iron metabolism, ferroptosis is shown to play an important role in HCC carcinogenesis and, more importantly, may hold the potential to eradicate HCC. In this review, we summarize the current knowledge we have of the role of ferroptosis in HCC and the application of ferroptosis as a therapy option and provide an overview of the potential translation of ferroptosis in the clinical practice of HCC.

## 1. Introduction

Historically, the term cell death comprised apoptosis, necrosis and autophagy, each possessing its own characteristic features, and—in case the of apoptosis and autophagy—specific molecular machineries and genetic programs [1,2]. Among these ‘classical’ forms of cell death, apoptosis was long considered to be the only form of regulated cell death (RCD), which made it an attractive target for pharmacological intervention by anticancer drugs [3]. In 2012, Dixon et al. described a new form of regulated cell death that functions independently of the apoptotic machinery and possesses its own morphological, biochemical and molecular characteristics [4,5]. As this new form of RCD is dependent on iron (accumulation) and oxidative stress-induced lipid peroxidation, it was termed ferroptosis [5,6]. Interestingly, earlier studies already found that treatment of cells with the substances erastin and RSL (which are now defined as class I and class II ferroptosis inducers) led to non-apoptotic cell death events [7,8]; however, Dixon et al. were the first to describe the underlying process of ferroptosis [4]. Due to the endpoint of ferroptosis characterized by a loss of plasma membrane integrity, its rupture and release of intracellular material, ferroptosis is currently considered a form of regulated necrosis [9,10] with a necrotic (lytic) morphology including typical mitochondrial changes (such as shrinkage, reduced cristae) [5,11]. Figure 1A indicates the position of ferroptosis in the currently defined spectrum of cell death mechanisms. 

The biochemical endpoint of ferroptosis, i.e., unrestricted/uncontrolled and severe lipid peroxidation, is reached via two major pathways (Figure 1B) [9,12]: first, the intrinsic or enzyme-regulated pathway relies on the inhibition of the activity of intracellular antioxidant systems—in the case of ferroptosis, especially the glutathione-dependent GPX4 system (glutathione peroxidase 4); second, the extrinsic or transporter-dependent pathway involves the (de-)regulation of amino acid transporters (system x_c_^-^, which imports cystine as a glutathione (GSH) precursor) or altered iron transport (via transferrin) or both, leading to an increase in the intracellular (labile) iron pool. From an experimental/methodological point of view, an operational definition of ferroptosis [13] involves the following requirements: cell death (by ferroptosis) is suppressed by both iron depletion (e.g., by ferrostatin 1) and lipophilic radical-trapping antioxidants (liproxstatin-1, vitamin E). Additionally, the direct detection of lipid peroxidation could be considered as further proof of ferroptosis induction [13]. 

Several requirements for ferroptosis and its biochemical hallmarks have been identified [5,6,9,13]: (i) supply of fatty acids (FA) as the substrate for later lipid peroxidation, (ii) ROS including free radicals, (iii) accumulation of free intracellular iron as a mediator of ROS generation and lipid peroxidation and (iv) a reduced activity of cellular antioxidant mechanisms. The latter can be summarized as three major ferroptosis-relevant ‘antioxidant axes’ [14], i.e., the cyst(e)ine/GSH/GPX4 axis, the GCH1 (GTP cyclohydrolase I)/BH_4_ (Tetrahydrobiopterin)/DHFR (Dyhydrofolate reductase) axis and the FSP1 (ferroptosis suppressor protein)/CoQ_10_ (Coenzyme Q10) axis (for details including potential pharmacological targets and drugs, see below). As illustrated in Figure 1C, these prerequisites together result in lipid peroxidation that, in an uncontrolled, severe, unrestricted manner, constitute the defining hallmark of ferroptosis [9,13]. The following paragraphs (see also Figure 2) explain the specific relevance of these biochemical hallmarks in the process of ferroptosis and mention the central molecular factors some of which, in turn, govern the sensitivity of cells towards ferroptosis induction. 

Requirement 1: fatty acids supply. PUFA represent good substrates for autoxidation [15] and the enzymes catalyzing the synthesis of PUFA-containing phospholipids ACSL4 (acyl-CoA synthetase long-chain family member 4) and LPCAT3 (lysophosphatidylcholine acyltransferase 3) represent factors that sensitize cells towards ferroptosis induction [13]. A deficiency of ACSL4 or LPCAT3 results in fewer targets of lipid peroxidation and, therefore, desensitizes/rescues cells from ferroptotic cell death [16,17,18]. The processes involved in the synthesis of PUFA-containing phospholipids as the central components of ferroptosis-related lipid peroxidation have been recently reviewed by Lee et al. [19].

Requirement 2: ROS and radicals. As reviewed recently by Sassetti et al. [20], the main source of intracellular ROS are mitochondria, i.e., incomplete reduction within the mitochondrial respiratory chain (superoxide). Furthermore, enzymes in the endoplasmatic reticulum, peroxisomes and the plasma membrane (such as NOX, NADPH oxidases) as well as non-enzymatically via the Fenton reaction represent sources of ROS [9,20] that might represent the starting point of ferroptosis induction—provided that the other mentioned requirements are fulfilled. 

Requirement 3: iron. The eponymous dependency of ferroptosis on iron is mirrored by several observations. Ferroptosis induced by specific triggers (erastin, cystine deprivation) can be blocked by the inactivation of the transferrin receptor (TfR1), which mediates cellular iron absorption transports (e.g., [21]). Intracellular iron not used for synthesis of Fe-containing enzymes is stored as ferritin—consequently inhibiting Fe storage in ferritin sensitizes towards ferroptosis (e.g., [22,23]) and reducing the autophagic liberation of iron from ferritin (ferritinophagy) increases cellular resistance towards ferroptosis (e.g., [24,25]). A similar correlation, i.e., a direct relationship between the intracellular labile iron (pool) and susceptibility towards ferroptosis, applies to the observations that cells become sensitive to ferroptosis induction by the inhibition of mechanisms that reduce intracellularly available iron—these include ferroportin 1 (FPN1, plasma membrane ferrous iron exporter) (e.g., [26,27]), the excretion of ferritin-bound iron via exosomes [28] and the transfer of iron to ferritin or other Fe-containing proteins via PCBP1 (poly-(rC)-binding protein 1) [29]. Functionally, iron acts as a mediator of ROS generation and subsequent lipid peroxidation [13,15]: first, the non-enzymatic, iron-dependent Fenton-like reactions amplify lipid peroxidation by the generation of the free radicals PLO^∙^ (alkoxyl phospholipid radical) and PLOO^∙^ (peroxyl phospholipid radical) after reaction with PLOOHs (phospholipid hydroperoxide). Second, enzymes involved in lipid peroxidation such as LOX (lipoxygenases) and POR (cytochrome P450 oxidoreductase) require iron for their catalytic reaction [13,14].

**Figure 2 cancers-14-01826-f002:**
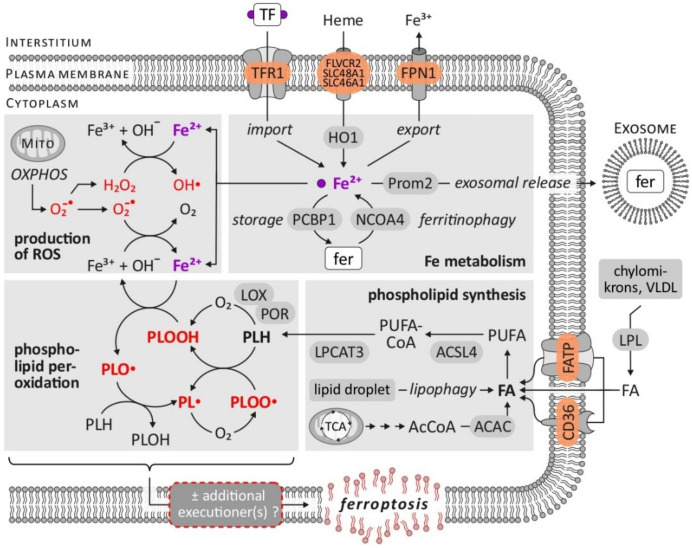
Ferroptosis—biochemical hallmarks and prerequisites. The figure summarizes the central biochemical characteristics of ferroptosis, i.e., generation of ROS, aspects of cellular iron metabolism, synthesis of phospholipids as targets of lipid peroxidation, as well as lipid peroxidation itself. While the latter represents the defining hallmark of ferroptosis, possible downstream mechanisms (executioner(s)) causing the final ferroptotic phenotype remain elusive. Abbreviations: ACAC = acetyl-CoA carboxylase, AcCoA = acetyl-coenzyme A, ACSL4 = acyl-CoA synthetase long-chain family member 4, FA = fatty acid, FATP = fatty acid transport protein, fer = ferritin, FLVCR2 = feline leukemia virus subgroup C cellular receptor 2, FPN1 = ferroportin 1, HO1 = heme oxygenase 1, LOX = lipoxygenase, LPCAT3 = lysophosphatidylcholine acyltransferase 3, LPL = lipoprotein lipase, MITO = mitochondrion, NCOA4 = nuclear receptor coactivator 4, OXPHOS = oxidative phosphorylation, PCBP1 = poly-(rC)-binding protein 1, PL∙ = phospholipid radical, PLH = phospholipid, PLO∙ = alkoxyl phospholipid radical, PLOO∙ = peroxyl phospholipid radical, PLOOH = phospholipid hydroperoxide, POR = cytochrome P450 oxidoreductase, Prom2 = prominin 2, PUFA = polyunsaturated fatty acid, PUFA-CoA = PUFA-coenzyme A, ROS = reactive oxygen species, SLC46A1 = Solute Carrier Family 46 Member 1, SLC48A1 = Solute Carrier Family 48 Member 1, TCA = tricarboxylic acid, TF = transferrin, TFR1 = transferrin receptor 1, VLDL = very low density lipoprotein. Based on: [12,14,19,30,31,32].

Requirement 4: reduced antioxidant mechanisms. Efficient induction and execution of ferroptosis requires the reduced activity of protective, cellular antioxidant mechanisms. As mentioned above, the three central axes of anti-ferroptosis regulation include the cyst(e)ine/GSH/GPX4 axis, the GCH1/BH_4_/DHFR axis and the FSP1/CoQ_10_ axis. The requirement of reduced efficiency of these systems for ferroptosis is demonstrated by the fact that several small-molecular weight inducers of ferroptosis specifically target and inactivate components of these antioxidant systems. Notably, so called class I ferroptosis inducers (FINs) inhibit system x_c_^-^ (and, consequently, limit cyst(e)ine availability and deplete GSH), class II FINs inhibit the downstream GPX4 antioxidant enzyme (and, consequently, lipid peroxide detoxification), and class III FINs deplete GPX4 and CoQ_10_ [3,33]. These classes I-III FINs might be referred to as ‘canonical’ inducers of ferroptosis while class IV FINs, which increase the labile iron pool, represent non-canonical inducers of ferroptosis [3]. 

Figure 3 summarizes the currently known pathways involved in inhibiting or promoting ferroptosis including established and experimental drugs that modulate these mechanisms. A specific discussion of experimental approaches targeting ferroptosis in HCC including the classes of drugs that have previously been investigated in the context of HCC is provided below. 

Interestingly, the eventual executioner(s) and the endstage effector(s) of ferroptosis currently remain uncertain [3,9,13]: continued oxidation of PUFAs might directly change membrane fluidity and permeability, cause thinning of the plasma membrane, increase its curvature and thus increase accessibility by oxidants [34]. Other downstream lipid peroxidation byproducts such as MDA (malondialdehyde) or 4-HNE (4-hydroxy-nonenal) may crosslink and inactivate proteins involved in essential cellular processes [35]. This assumption involving lipid peroxides as final executioners of ferroptosis is hampered by the lack of a quantifiable threshold of PLOOHs defining the endstage of ferroptosis, i.e., a point of no return [9]. As summarized by Tang and Kroemer [9], other hypotheses include either currently unknown pore-forming proteins activated by post-translational modifications or altered subcellular localization, the release of lysosomal hydrolases followed by the destruction of plasma membrane components or a deranged phospholipid bilayer (a)symmetry induced by flippases or scramblases that might compromise membrane function and integrity. As the molecular mechanisms of ferroptosis have been intensively investigated in recent years, ferroptosis has more and more become a possible and innovative therapeutic strategy against different human diseases especially liver diseases (including ischemia/reperfusion-related injury, nonalcoholic fatty and alcoholic liver diseases, hemochromatosis to drug-injured liver injury) and in HCC as reviewed recently in detail [1,33,36,37]. 

## 2. The Potential of Ferroptosis in (Hepatocellular) Cancer Therapy

### 2.1. Ferroptosis in Cancer

Inducing cell death in cancer is an obvious and common therapeutic strategy for combating malignant diseases. One mechanism, which leads to cancer cell death by chemotherapeutics, such as cisplatin, is apoptosis, the most well-known form of regulated cell death [38]. Apoptosis, also known as programmed cell death, is characterized by nuclear and cellular volume reduction, nuclear fragmentation and the blebbing of the plasma membrane and can be induced by DNA damage—one main mechanism of chemotherapeutics [39,40]. Although inducing apoptosis has shown great success in cancer treatment, several cancer entities frequently develop resistance towards apoptosis-inducing substances and finding an alternative therapy becomes an urgent necessity. One potential novel treatment strategy would be the induction of ferroptosis, an apoptosis-independent form of cell death [41]. One particular characteristic of cancer, which might be a reason for ferroptosis sensitivity, is the fact that cancer cells usually have a high(er) demand for iron compared to normal cells [42]. This increased iron uptake is needed for aberrant proliferation and metabolism of tumor cells and results in an elevated oxidative stress level [42,43]. To compensate the high oxidative stress, cancer cells tend to activate and upregulate the transcription and expression of antioxidant factors and genes, such as GPX4, SLC7A11 (Solute Carrier Family 7 Member 11) and FSP-1, which have been described as central ferroptosis regulators in the previous chapter [42,43,44]. Therefore, to induce ferroptosis, it is necessary to prevent the cancer cells to compensate the elevated oxidative stress by inhibiting/degrading the antioxidant protection systems (GPX4, FSP1) and to increase the intracellular iron level [42,43]. Interestingly, current studies suggest that ferroptosis can be induced in several cancer cell lines and xenograft models to inhibit tumor growth and proliferation [45,46]. Cancer cell lines from pancreatic cancer, renal cell carcinoma, ovarian cancer, breast cancer, diffuse large B-cell lymphoma (DLBCL), lung carcinoma, meningioma, prostate cancer, rhabdomyosarcoma, osteosarcoma, as well as cervical cancer, displayed a susceptibility towards ferroptosis-inducing substances [45,46]. In 2014, Yang et al. demonstrated, that the growth of diffuse large B-cell lymphoma cell lines was attenuated by treatment with ferroptosis-inducing substances [47]. Additionally, these substances enabled lipid peroxidation and the decline in cell viability of DLBCL cells could be rescued via vitamin E, an antioxidant [47]. These results suggest that a ROS-dependent induced ferroptotic cell death can be readily induced in cells derived from various cancer entities. Although such preclinical data suggest a promising anticancer approach via the induction of ferroptosis, clinical studies are currently not available and the usage of ferroptosis inducing substances need to be further studied in situ [43,48]. Preclinical data regarding hepatocellular carcinoma demonstrate a promising outlook to combat liver cancer with the induction of ferroptosis [49,50]. The following chapter discusses the conceptual background why ferroptosis might be especially attractive for novel approaches in HCC treatment. 

### 2.2. The Potential Role of Ferroptosis in Hepatocytes and Liver Pathologies

#### 2.2.1. Hepatocellular Carcinoma

HCC is a deadly malignancy and the most prevalent form of human liver cancer [51]. HCC was responsible for over 830,000 cancer related deaths worldwide in 2020 [51]. Therefore, HCC is the second most leading cause of cancer deaths around the globe [51]. In general, patients are often diagnosed at an already mid-to-late stage, preventing surgical treatment due to unspecific symptoms, such as nausea, vomiting, fatigue, abdominal pain and unintentional weight loss [50]. Treatment options for advanced-staged patients include post-surgery adjuvant chemotherapy, immunotherapy and targeted therapy with sorafenib and regorafenib, if sorafenib treatment is unsuccessful [52,53]. Sorafenib is a multikinase inhibitor that inhibits downstream intracellular threonine/serine kinases like RAF/BRAF (rapidly accelerated fibrosarcoma) and tyrosine kinase receptors that are located on the cell surface (such as vascular endothelial growth factor (VEGF) and platelet-derived growth factor (PDGF)) [54]. These kinases are involved in tumor angiogenesis and tumor cell growth and blocking them with sorafenib induces apoptosis in HCC [55,56,57]. Clinically, sorafenib can enhance the survival of HCC patients [54]. However, several studies revealed a general resistance of HCC against sorafenib treatment, thus calling for alternative therapies, other than apoptosis, that can induce or lead to alternative cell death forms in cancer [58]. 

#### 2.2.2. Iron in the Liver

In general, the liver plays a central role regarding iron homeostasis in the human body. It produces several proteins that are required for the maintenance of iron homeostasis, it provides mobilized iron for metabolic requirements throughout the whole body and an excess of iron is mostly stored in the liver [42]. The uptake of iron in the liver occurs mostly via the transferrin–transferrin receptor pathway under normal conditions [42]. Iron that is in the plasma, is bound to transferrin, a plasma glycoprotein that can sequester metal irons safely [42]. The iron-loaded transferrin bounds then to TfR and the iron-transferrin-complex is transported into the cytosol via a receptor-mediated endocytosis pathway [59]. Under iron overload conditions, the non-transferrin-bound iron (NTBI) pathway is activated, because iron-uptake by plasma transferrin is not sufficient enough [42]. In this case, NTBI is transported via divalent metal transporter 1 (DNMT1) or ZRT and IRT-like protein 14 (ZIP14) from the plasma into the liver [42]. Once the iron is in the cytosol, PCBP1 and PCBP2 are responsible for the delivery of iron to ferritin and, because of the PCBPs chaperon activities, iron toxicity can be restricted to a certain extent in the cytosol [42]. 

Upto 13–15 mg of iron/g dry weight of liver are considered normal and are stored in the hepatocytes, the main cellular unit of the liver [60]. Here, the main storage location of iron are the cores of ferritin shells [42]. These ferritin shells consists of a hollow protein named apoferritin [42]. Apoferritin surrounds an aqueous hollow space where up to 4500 iron atoms can be stored [42]. Apoferritin consists of a H-subunit, which is responsible for the detoxification of iron and the L-subunit, which contributes to the long term storage and the mineralization of iron [42]. 

#### 2.2.3. Ferroptosis/Iron Homeostasis in Hepatic Diseases

As the liver plays a crucial role in iron homeostasis, slight changes and unregulated free iron, the labile iron pool, might contribute to hepatic damages and diseases [61,62]. Therefore, the ferroptosis that is associated with iron overload, is involved in the progression of different kinds of hepatic illnesses and plays a dual role in the liver [61,62]. Ferroptosis-related hepatic pathologies encompass acute liver diseases (ferroptosis-induced liver injury) and chronic liver diseases (Hepatitis B/C, non-alcoholic fatty liver diseases (NAFLD) and hereditary hemochromatosis), which might finally progress to HCC [61]. For example, in hereditary hemochromatosis that comprise a heterogeneous cohort of inherited iron overload diseases, particularly caused by the mutation of proteins that restrict iron absorption (homeostatic iron regulator (HFE), hemojuvelin (HJV), FPN1), excessive iron is absorbed by organs such as the heart, the intestine and the liver [63]. The permanent accumulation of iron might lead to liver tissue damage that may ultimately result in HCC [42]. In 2017, Wang et al. showed that iron overload and ferroptotic processes were involved in hereditary hemochromatosis [64]. In an established hereditary hemochromatosis mouse model that lacks the HFE and HJV genes, they observed that these mice displayed iron overload and increased lipid peroxidation compared to wild-type mice [64]. To further confirm that ferroptosis was involved, mice were treated with a ferroptosis inhibitor, ferrostatin-1, which attenuated hereditary hemochromatosis-induced liver damage [64]. 

Another chronic liver disease that is associated with iron overload and ferroptosis is NAFLD [65]. The NAFLD subtype NASH, non-alcoholic steatohepatitis, leads to obesity and metabolic syndrome [65]. Interestingly, NASH is associated with elevated lipid peroxidation and iron accumulation in hepatocytes leading to inflammation and hepatic cell death [66]. Vitamin E for example, decreased serum transaminase levels and lipid peroxidation in NASH patients and furthermore, ferroptosis inhibitors such as deferiprone and trolox suppressed inflammation and hepatic cell death at the initial stage of NASH [61]. These findings corroborate the role of ferroptosis in several liver diseases which might represent risk factors for the subsequent development of HCC.

#### 2.2.4. Iron and PUFAs in HCC

In HCC, ferroptosis might play a different role compared to other forms of liver diseases. Several studies demonstrated that iron overload in HCC is a driver of tumorigenesis, proliferation and tumor growth [49,67]. Therefore, one approach of anticancer therapy might be the maintenance or increase of abnormal iron levels in HCC cells in order to induce overproduction of ROS resulting in ferroptotic cell death. Another factor that might contribute to the putative susceptibility of HCC cells towards ferroptosis are PUFAs. Lim et al. demonstrated that PUFAs can reduce HCC growth via the inhibition of β-catenin and cyclooxygenase-2 (COX-2)—two factors that promote tumorigenesis in HCC [68]. Additionally, Weylandt et al. demonstrated that PUFAs consumed in the form of fish that contain unsaturated fatty acids (n-3 PUFA) can reduce the risk of HCC development [69]. Therefore, the observed iron overload and the antitumor effect of PUFAs in HCC, two key players needed to produce lipid ROS in ferroptosis, indicate why HCC might be an attractive target for ferroptosis-based therapies.

#### 2.2.5. GPX4 and SLC7A11 Expression in HCC

Furthermore, several studies demonstrated that the ferroptosis regulators GPX4 and SLC7A11 are overexpressed in HCC compared to normal cells/tissues [70,71]. Guo et al. showed that the overexpressed SLC7A11 in HCC is often associated with tumor progression as well as poor prognosis, whereas SLC7A11 suppression attenuated HCC cell proliferation [70]. In addition, GPX4 was also significantly overexpressed and associated with a higher malignant grade in HCC patients demonstrated by Guerriero et al. [71]. As these factors are important for eliminating oxidative stress in (cancer) metabolism and cell growth and are evidentially overexpressed in HCC, ferroptosis induction via their pharmacological inhibition, degradation or both might be a promising anti-HCC strategy. The following chapter summarizes the currently known molecular effects and mechanisms in HCC cells treated with ferroptosis-inducing compounds.

## 3. The Pharmacological Induction of Ferroptosis in HCC Cells

As mentioned in the introduction, induction of ferroptosis requires inhibition or down-regulation of mechanisms that protect cells from the accumulation of lipid peroxides to lethal levels [4]. Based on the different mechanisms of ferroptosis induction, four classes (I to IV) of FINs are currently defined [41]. 

The glutathione system as well as GPX4 are major mechanisms that counteract lipid peroxidation and thereby inhibit ferroptosis [4,47]. Reduced GSH is an important antioxidant that is converted to oxidized glutathione (GSSG) by GPX4 during the reduction of lipid radicals [47]. As summarized in Figure 3, GSH is generated from glutamate, cysteine and glycine in a two-step biosynthetic pathway. Consequently, cysteine availability limits the synthesis of glutathione. System x_c_^-^ is an amino acid antiporter system located at the plasma membrane that imports cystine in exchange for intracellular glutamate [72]. In the cell, cystine is converted to cysteine and then used for synthesis of glutathione [72]. System x_c_^-^ is a heterodimeric amino acid transporter composed of a SLC7A11 and a SLC3A2 (Solute Carrier Familiy 3 Member 2) subunit, where the SLC7A11 subunit is responsible for the actual transport activity [72]. It was shown that the inhibition of system x_c_^-^ and specifically the SLC7A11 subunit resulted in reduced cysteine and glutathione levels and the induction of ferroptosis [4,73]. Substances that induce ferroptosis via inhibition of system x_c_^-^ are categorized as class I FINs [3]. Amongst them, erastin is the most frequently used class I FIN [3]. However, (cancer) cells show different sensitivity towards ferroptosis induction via the inhibition of system x_c_^-^, as the system x_c_^-^ can be bypassed by the transsulfuration pathway synthesizing cysteine from methionine to supply glutathione biosynthesis [74]. 

The second main strategy to induce ferroptosis in cells is downstream of system x_c_^-^ via the inhibition or degradation of GPX4 [4,47]. As GPX4 uses GSH to reduce lipid radicals, the manipulation of GPX4 activity results in the accumulation of lipid peroxides to lethal doses and the induction of ferroptosis [47]. Class II FINs directly inhibit GPX4 and thereby induce ferroptotic cell death [47]. RSL-3 is the prototype class II FIN and several studies described a potent cytotoxic effect in tumor cells [4,75]. FIN56 is a compound that was discovered during a large screen for ferroptosis-inducing substances [76]. FIN56 also induces ferroptosis by interaction with GPX4; however, in contrast to RSL-3, FIN56 causes GPX4 protein degradation and is therefore classified as a class III FIN [76]. The prototype of class IV FINs is FINO2, a substance that has two ferroptosis-inducing effects as it indirectly inhibits GPX4 and modifies the labile iron pool [77]. 

As discussed in chapter 2 and as HCC shows a high resistance to commonly used therapeutic approaches, the induction of ferroptosis as a new therapeutic strategy appears attractive. In fact, in recent years, several reports demonstrated the substance-based induction of both in vitro and in vivo ferroptosis in HCC, using different drugs (see Table 1). 

Sorafenib is a multikinase inhibitor used for the treatment of advanced HCC [52]. In 2013, Louandre et al. found that sorafenib induces a form of cell death that showed no typical signs of apoptosis but rather characteristics of ferroptosis [81]. Likewise, other studies also described that sorafenib induces ferroptosis in various HCC models in vitro and in vivo [79,80,82]. Mechanistically, sorafenib acts as an inhibitor of system x_c_^-^ and can therefore be categorized as a class I FIN [78]. As kinase inhibitors with similar kinase inhibition patterns were not able to induce ferroptosis, the authors postulated that sorafenib causes ferroptosis via the indirect inhibition of system x_c_^-^ [78]. Recently, another publication contributed to a better understanding of the mode of ferroptosis induction in HCC cells by sorafenib. The authors found that sorafenib treatment for 60 min resulted in changes at phosphorylation sites of proteins that are involved in iron homeostasis and of proteins that are part of ferroptosis-related pathways [83]. However, there are also reports that sorafenib fails to trigger ferroptosis in different cancer cell lines including HCC [93], which might indicate cell line-specific effects, other (molecular) factors that contribute to the sensitivity of sorafenib-induced ferroptosis or both. In this regard, Louandre at al. showed that HCC cells with reduced levels of the retinoblastoma (Rb) protein were more vulnerable to sorafenib-induced ferroptosis [82]. Moreover, sorafenib caused complete tumor regression (in 50% of tested cases) in a xenograft model using HCC cells with RNA-mediated silenced Rb protein, whereas no such effect was seen in the control group [82]. The nuclear factor erythroid 2-related factor 2 (NRF2) is a key regulator of antioxidant response and a critical mitigator of lipid peroxidation and ferroptosis by targeting genes that are involved in preventing the formation of lipid peroxides including GPX4 [94]. NRF2 was identified to determine the sensitivity of HCC cells to ferroptosis following sorafenib treatment [95]. It was observed that upon induction of ferroptosis, NRF2 protein levels were stabilized which in turn causes its nuclear translocation and activation of genes involved in the antioxidant response. Consequently, the silencing of NRF2 sensitized HCC cells towards ferroptosis induction by sorafenib resulting in significant anticancer effects in vitro and in vivo [95]. Recently, it has been demonstrated that hyperactivation of NRF2 leads to increased cellular glutamine dependency and that NRF2 antioxidant pathways are activated by mutant KRAS, both of which contribute to chemoresistance. In pancreatic cancer, the oncogenic KRAS driven upregulation of NRF2 modulates GPX4 transcription and glutamine metabolism pathways, which are in favor of glutaminolysis. Interference with these pathways restrains stress granule assembly (an indicator of chemoresistance), and glutamine deprivation of the cells reduces GPX4 levels. These interrelations suggest that KRAS-mutant cancer cells that display high levels of glutaminolysis are more susceptible to ferroptosis. Moreover, modulating NRF2 expression regulates the sensitivity of pancreatic cancer cells to gemcitabine. The inhibition of glutaminase sensitizes the cancer cells towards gemcitabine and thus improves the effectiveness of chemotherapy. Hence, targeting glutamine metabolism turns out to be an efficient strategy in overcoming chemoresistance. As such, the first-in-class small molecule glutaminase inhibitor, CB 839, improves gemcitabine efficacy, and the combination of the drugs enhances tumor regression in a mouse pancreatic cancer model [96]. Inhibition of NRF2 by the cellular pro-oxidant quiescin sulfhydryl oxidase 1 (QSOX1) also promoted ferroptotic cell death in HCC cells [84]. Using an in vivo model, treatment with sorafenib resulted in a pronounced reduction of the tumor volume caused by HCC cells that overexpressed QSOX1 [84]. There are also attempts to enhance the efficacy of sorafenib to induce ferroptosis in HCC cells, not only to increase the cytotoxic effects, but also to reduce the sorafenib concentrations required to induce ferroptosis—in order to reduce sorafenib-related side effects. Li and coworkers combined artesunate with sorafenib to induce ferroptosis in HCC cells [85]: a combination of artesunate and low doses of sorafenib effectively reduced HCC cell viability accompanied with significant accumulation of lipid ROS. Mechanistically, they found that combined artesunate and sorafenib treatment causes ferritin degradation and the subsequent release of iron, a process termed ferritinophagy [85].

Another link between ferroptosis and the use of sorafenib to treat HCC is provided by cytoplasmatic stress granules (SG): SG form during chronic liver diseases as well as HCC and, in general, they form as a response to cellular stresses such as nutrient shortage, hypoxia, oxidative and ER stress among others [97]. Importantly, several anti-cancer treatments including sorafenib trigger the assembly of SG and might confer the resistance of (HCC) cancer cells towards these therapies [97,98]. A central component of SGs, the Ras-GTPase-activating protein SH3 domain-binding protein 1 (G3BP1), can be bound by the long noncoding RNA P53RRA which, in turn, leads to the higher retention of p53 in the cell nucleus accompanied by increased ferroptosis [99]. Another association between ferroptosis and SGs is based on the ELAVL1/HuR protein ((embryonic lethal, abnormal vision, Drosophila)-like 1, Hu-antigen R) which accumulates in SGs during cellular stresses and might confer a cell survival advantage via the stabilization of several oncogenic transcripts (for details, see e.g., [100]). Interestingly, while HuR expression correlates with a poor clinical outcome in HCC [101], it additionally promotes ferroptosis induced by sorafenib [102]; however, a causal link between sorafenib resistance and SG formation remains to be investigated [100]. 

Glutathione S-transferase zeta 1 (GSTZ1), the penultimate enzyme in tyrosine and phenylalanine catabolism, might influence sorafenib resistance negatively in HCC via NRF2 [103]. In HCC, GSTZ1 is suggested to act as a tumor suppressor, because of its observed downregulation, which results in poorer prognosis for patients and enhanced carcinogenesis [104]. Recent studies demonstrate, that GSTZ1 deficiency in HCC can activate NRF2-associated pathways [103,104,105]. Li et al. could show that GSTZ1 downregulation resulted in depleted GSH and increased ROS levels, which led to the activation of NRF2-associated pathways [105]. In another study, carried out by Yang et al., the NRF2/IGF1R (insulin-like growth factor)-associated pathway was activated by the accumulation of succinylacetone, a carcinogenic metabolite, induced by the GSTZ1 deficiency, that leads to the inactivation of kelch-like ECH-associated protein 1 (KEAP1), a negative regulator of NRF2 [104]. In sorafenib-resistant HCC cells, Wang et al. could demonstrate that GTSZ1 was downregulated, and this observation was associated with NRF2 pathway activation and increased GPX4 levels [103]. The overexpression of GSTZ1 led to an enhanced sensitivity of HCC cells towards sorafenib-induced ferroptosis [103]. Furthermore, combinatory treatment of GSTZ1 deficient HCC cells with sorafenib and RSL-3, led to a decrease of cell viability and ferroptosis promotion [103]. These results indicate that GSTZ1 harbors an important role in sorafenib resistance.

Alongside sorafenib as a class I FIN, erastin was also shown to cause ferroptosis in HCC cells [79,95]. Interestingly, also for erastin, combinatory treatments resulted in synergistic effects regarding the induction of ferroptosis. Lippmann et al. observed that a combination of erastin and buthionine sulfoximine (BSO) resulted in significantly reduced cell viability in HCC cells and that this effect was reversible by ferroptosis inhibitors [86]. 

Although the investigation of ferroptosis induction in HCC cells was mostly done using class I FINs, there are also reports regarding other FIN classes and substances. Asperti et al. described a strong ferroptotic effect of RSL-3 in HepG2 cells [106]. Similarly, Zhao and coworkers observed strong cytotoxicity of RSL-3 in HCC cells [87]. Interestingly, this effect was dependent on the extracellular lactate levels, as high lactate levels caused resistance of HCC cells towards RSL-3-induced ferroptosis [87]. As described before, RSL-3 causes ferroptosis by the direct inhibition of GPX4, indicating that tumor cells that rely heavily on GPX4 might be especially vulnerable to class II FIN treatment. The potential of this approach is underlined by the fact that GPX4 is overexpressed in HCC tissues and that GPX4 expression was higher in grade III HCC tissues compared to lower graded tumors [71].

The potential of ferroptosis as an apoptosis-independent anticancer strategy led to the identification of ferroptosis-inducing substances distinct from the established four FIN classes. For example, via compound screening, the natural compound saponin formosanin C was found to cause ferroptosis in HCC cells by the induction of ferritinophagy [88]. Of note, compared to sorafenib, the toxic effect of saponin formosanin C was much stronger [88]. Another natural product, heteronemin, was found to induce both apoptosis and ferroptosis in HCC cells [89]. The authors observed that the apoptosis inhibitor Z-VAD-FMK only reversed about 20% of the cytotoxic effect of heteronemin and subsequently found that treatment with heteronemin also resulted in cell death with typical traits of ferroptosis [89]. Yet another natural compound, solasonine, was identified to trigger ferroptosis in HCC cells [90]. As mentioned before, targeting GPX4 to trigger ferroptosis might be a promising approach for HCC. Interestingly, Jin et al. found that solasonine causes ferroptosis by suppression of GPX4. Moreover, solasonine also reduced the migratory and invasive potential of HCC cells in vitro [90]. 

NUPR1 (Nuclear Protein 1) is a stress-inducible protein that is expressed in most cancer cells and is involved in several cellular processes including the regulation of cell cycle and therapeutic resistance. Furthermore, it was shown that NUPR1 is activated in response to induction of ferroptosis and acts as a repressor of ferroptosis [107]. In a recent study, the authors investigated the effect of the NUPR1 inhibitor ZZW-115 in HCC cells and found that ZZW-115 caused ferroptosis based on the induction of an imbalance of the GSH/GSSG ratio and the inhibition of GPX4 activity [91]. In addition, the authors demonstrated that the addition of ZZW-115 increased the effect of the established FINs erastin and RSL-3. In vivo, ZZW-115 also reduced GPX4 activity as well as GPX4 mRNA levels [91]. 

As mentioned before, NRF2 protects cells from ferroptotic cell death. Bai et al. identified sigma-1 receptor (S1R) as a blocker of ferroptosis via the involvement of NRF2 [92,108]. They found elevated S1R protein levels following the treatment of HCC cells with sorafenib and that the inhibition of NRF2 causes the overexpression of S1R mRNA [92]. Pharmacological inhibition of S1R using haloperidol significantly increased the sensitivity of HCC cells towards sorafenib-induced ferroptosis [92].

Drug delivery via nanoparticles represents a relatively young but promising approach for target-oriented and site-specific therapy. Tang et al. loaded manganese-silica nanoparticles with sorafenib and applied this formulation to HCC cells [109]. They found that the degradation of the manganese-silica nanoparticles itself (via the cleavage of -Mn-O- bonds) was able to induce ferroptosis through consumption of GSH. Combined with the effects of the released sorafenib, this strategy yielded strong antitumor effects [109]. In a different approach, low-density lipoprotein nanoparticles were loaded with the natural omega-3 fatty acid docosahexaenoic acid, LDL-DHA [110]. The treatment of HCC cells with LDL-DHA resulted in accumulation of lipid ROS and GPX4 inactivation in vitro as well as ferroptotic cell death in vitro and in vivo [110]. Mechanistically, besides the inactivation of GPX4, LDL-DHA causes the accumulation of GSSG and NADP+ and thus increases the susceptibility to ROS formation and oxidative damage [110]. Engineered exosomes can also be used as carriers of drugs. Du and coworkers loaded exosomes with erastin and a photosensitizer using sonication and used these exosomes to treat HCC cells in vitro and in vivo. Following the administration of these engineered exosomes, irradiation led to increased lipid ROS levels and ferroptosis [111].

## 4. The Translation of Ferroptosis into Clinical HCC Practice

This chapter deals with the issue of how the basic knowledge of (pharmacologically) induced ferroptosis could be translated into “daily” clinical practice for HCC patients. In particular, possible therapeutic scenarios for HCC treatment based on the combination of ferroptosis induction with standard chemotherapy, targeted and immunotherapy as well as local ablative techniques and radiotherapy will be presented on the basis of “pre-clinical” experiments or first clinical trials.

### 4.1. Ferroptosis Scoring System

The first and more theoretical approach for integrating ferroptosis in the difficult process of optimizing HCC treatment modalities was to develop a predictive and prognostic ferroptosis scoring system (see Table 2 for an overview). Besides combinatory treatment strategies, the in silico analysis of ferroptosis-related genes in HCC seems to be an interesting approach: Gao et al. developed a scoring model based on these genes for prognosis and immunotherapy response prediction and tumor microenvironment evaluation in HCC samples derived from TCGA (The Cancer Genome Atlas, *n* = 377) and GEO databases (Gene Expression Omnibus, *n* = 115) [112]. This study demonstrated that (i) the ferroptosis gene cluster, called Ferrcluster B, and a high ferroscore group is linked to lower overall survival and that (ii) a high ferroscore group classification is associated with the specific in situ expression pattern of programmed death-ligand 1 (PD-1) and to the efficacy of immune checkpoint inhibitors against PD-1 or PD-1 plus cytotoxic T-lymphocyte associated protein 4 (CTLA4). 

Deng et al. could identify two ferroptosis activity-associated subtypes using transcriptome and methylome data from 374 HCC cases with 41 ferroptosis-related genes. Based on these findings, they designed and validated a 15-gene ferroptosis-related prognostic model (FPM) for HCC for accurate risk stratification in a second database with an additional 232 HCC cases from another independent cohort [113]. Patients with the so called “Ferroptosis-H” phenotype show worse overall and disease-specific survival, which is linked to specific molecular subtypes including mRNA expression patterns, tumor mutation profiles and micro-environmental immune status. 

Next, Liu and co-workers extracted and validated two heterogeneous ferroptosis subtypes from 74 ferroptosis related genes in 3933 HCC samples from 32 datasets, whereby the ferroptosis subtype “C1” was related to a lower metabolism and a higher immunity status as well as the opposite status for the ferroptosis subtype “C” [114]. 

Additionally, a comprehensive index of ferroptosis and immune status (CIFI) was constructed by combining data from FerrDb and ImmPort with datasets of the GEO GSE14520 (*n* = 220) and the TCGA (*n* = 365) database [115]. The authors demonstrated that the subgroup of patients with a high CIFI value had a worse prognosis linked to increased suppressors of ferroptosis paralleled by immunosuppressive cells like cancer-associated fibroblasts (CAFs) and myeloid-derived suppressor cells (MDSCs). The authors concluded and postulated that this CIFI has predictive and prognostic potency for the selection of patients for immunotherapies and targeted therapies.

Zi-An Chen et al. developed a predictive and prognostic ferroptosis-related signature model based on 2 ferroptosis-related mRNAs (SLC1A5 and SLC7A11) and 8 ferroptosis-related lncRNAs (AC245297.3, MYLK-AS1, NRAV, SREBF2- AS1, AL031985.3, ZFPM2-AS1, AC015908.3, MSC-AS1) in HCC [116]. The findings revealed differences of tumor microenvironment and immune cell infiltration as well as tumor-related pathways between low- and high-risk groups according to the established ferroptosis-related signature model in HCC. Interestingly, the authors could identify 10 significant candidate drugs by integrating in their findings in the L1000FWD database, which could be helpful for further experimental steps for targeting HCC.

In relation to HCC, Zhang et al. constructed a ferroptosis score based on detailed in silico analysis of three different databases with total of 174 cases to predict the efficacy and prognosis of patients with cholangiocarcinoma treated with photodynamic therapy [117]. Furthermore, the authors could verify and transfer their in-silico-findings by immunohistochemistry, western blot and RNA microarray analyses in vitro and in vivo indicating the reproducibility of such “theoretical” data. As ablative techniques like transarterial embolization (TAE), transarterial chemoembolization (TACE), transarterial radioembolization (TARE), radiofrequency or (RFA) and microwave ablation (MWA) that generate ROS in situ are also routinely applied to HCC, transferring ferroptosis scores could also impact on prediction/prognosis of such ablative techniques in HCC, although clinical validation is still pending [118].

Finally, the study of Ji Feng et al. could demonstrate that ACSL4, a ferroptosis-promoting enzyme, represents a predictive biomarker for sorafenib sensitivity in HCC in vitro and in vivo [119]. The investigation of expression of ACSL4 in HCC tumor specimens revealed that the high baseline expression of ACSL4 in untreated HCC tissue is related to complete or partial responses to sorafenib treatment in comparison to the HCC group with low ASCL4 expression [119]. 

In summary, the mentioned in silico analyses convincingly indicated that the expression of ferroptosis-associated genes should be integrated as a new predictive and prognostic biomarker in the established classification of HCC [120]. Patients with HCC could have the benefit of such ferroptosis-related sub-classification with regard to the emerging use of immune checkpoint inhibitors [118]—provided that such classifiers can be successfully clinically validated.

### 4.2. Nanoparticles and Exosomes

Next, a more active and more therapeutic approach using nanoparticles or exosomes to integrate ferroptosis into the HCC therapy concept is currently being explored. The basic concept is based on a double carrier model to transfer a ferroptosis inducer in combination with a chemotherapeutical or targeting drug to the cancer cells. 

Qiao-Mei Zhou et al. developed iron-doped nanoparticles containing doxorubicin. Doxorubicin and iron act synergistically on the induction of tumor cell death via ferroptosis and apoptosis [121]. Furthermore, this platform with a superparamagnetic framework could be used to monitor treatment under T2-weighted magnetic resonance imaging as well [121]. Therefore, the authors concluded that such a nanoplatform could integrate cancer diagnosis, treatment and the monitoring of HCC. 

Tang and colleagues designed a dual GSH-exhausting sorafenib loaded manganese-silica nanodrug for inducing ferroptosis in HCC cells via the consumption of intracellular GSH and the inhibition of intracellular GSH synthesis [109]. 

Xu et al. constructed a manganese porphyrin-based metallo-organic framework to be used as a nanosensitizer to self-supply oxygen (O_2_) and to decrease GSH for ultrasound-triggered sonodynamic therapy. The authors could show strong anticancer and anti-metastatic activity in an in vivo model with hepatocellular and breast carcinoma of the mouse (H22 and 4T1), which was interestingly paralleled by an immunosuppressive microenvironment through increased activated CD8+ T cells and decreased myeloid-derived suppressor cells in situ [122]. 

Another nanoparticle-based approach used a cascaded copper-based metallo–organic framework nanocatalyst which bears the cyclooxygenase-2 inhibitor meloxicam and the targeted agent sorafenib to amplify the efficiency of HCC therapy by ferroptosis [123]. 

Finally, Ou et al. chose LDL-DHA nanoparticles to induce ferroptotic related cell death in HCC [110]. Based on their experimental setting, they could convincingly demonstrate that LDL-DHA-treated HCC cell lines in vitro and tumors in vivo exhibited ferroptotic cell death through increased levels of tissue lipid hydroperoxides and the suppression of GPX4 expression [110]. 

Another interesting approach was performed by Do et al. who designed exosomes for targeted and efficient ferroptosis induction in cancer via chemo-photodynamic therapy [111]. In detail, the authors developed exosome donor cells (HEK293T) that were transfected with CD47-overexpressing plasmid and loaded with erastin and a photosensitizer (Rose Bengal, RB). These drug-loaded exosomes (Er/RB@ExosCD47) could significantly induce ferroptosis both in vitro and in vivo in tumor cells after laser irradiation at 532 nm without showing toxicity in normal liver [111]. 

Highly sophisticated nanoparticle-based platforms delivering ferroptosis-inducing drugs in combination with standard drugs for HCC seem to be a promising step towards integrating ferroptosis into the treatment “portfolio” of HCC in the future. 

### 4.3. Long Noncoding RNAs/miRNA

Another approach is based on the interaction of ferroptosis and either long noncoding RNAs (lncRNAs) or miRNA as identified via in silico [116,117,124] or via in vitro/in vivo analyses [125]. 

Looking in detail at sophisticated in silico techniques, Wang et al. could identify a ferroptosis-specific lncRNA signature, which could serve as an independent prognostic biomarker for the overall survival of patients with HCC. These five extracted lncRNAs (LUCAT1, AC099850.3, AL365203.2, AL031985.3, AC009005.1) could be linked to an HCC specific tumor microenvironment (especially dendritic cells (DCs), macrophages, mast cells, follicular helper T cells, Th1/2 cells, Th2 cells and regulatory T cells) and to the anti-cancer ability of immune checkpoint inhibitors to predict the response to immunotherapy in HCC [126]. 

Next, Huang et al. applied various bioinformatics methods to crystallize an immune- and ferroptosis-related lncRNA signature for the prognosis of HCC based on the following 17 candidate LncRNAs after filtration: AC009005.1, AC016773.1, AC090164.2, AC092119.2, AC099850.3, AL021807.1, AL356234.2, AL359510.2, CASC9, DUXAP8, GDNF-AS1, LINC01224, LINC01436, LINC02202, LUCAT1, PTGES2-AS1, and ZFPM2-AS1 [124]. 

The application of the designed immune- and ferroptosis-related (IF) lncRNAs signature (finally on eight lncRNAs) predicts a worse outcome in patients with HCC that have a high IFlncRNA signature in comparison with those with a low IFlncRNA signature [124]. Comparable to the in-silico results of Wang et al. the IFlncRNA signature could be correlated to inflammatory cell infiltrates and the expression of immune checkpoints highlighting the potential predictive potency of such an IFlncRNA signature for the response to immune checkpoint inhibitor treatments for HCC. Interestingly, the comparison of the half maximal inhibitory concentration (IC50) of 30 anti-tumor drugs on patients with HCC and an IFlncRNA signature revealed that patients with high IFLSig should show no benefit from gefitinib, mitomycin, temsirolimus and erlotinib on the one hand, but a possible benefit from bexarotene, metformin, sorafenib, bleomycin and lapatinib on the other hand [124]. Therefore, the authors of this study suggested that IFLSig could help for the precise selecting chemotherapeutic drug against HCC in relation to the possible clinical benefit.

As mentioned before, Zi-An Chen et al. could develop a Ferroptosis-related signature predictive and prognostic model based on 2 ferroptosis-related mRNAs (SLC1A5 and SLC7A11) and 8 ferroptosis-related lncRNAs (AC245297.3, MYLK-AS1, NRAV, SREBF2- AS1, AL031985.3, ZFPM2-AS1, AC015908.3, MSC-AS1) in HCC [116]. 

Integrating the three sets of lncRNA in a classical Venn diagram revealed the most overlap for AC009005.1, AC099850.3 and LUCAT1, which was shown to play a relevant role for autophagy [127] and could amplify ferroptosis by degradation of ferritin [128]. 

When looking at single miRNAs, it could be shown that the miRNA 214-3p (miR-214) plays a regulatory role in the hepatocarcinogenesis via the enhancement of erastin-induced ferroptosis and targeting activating transcription factor 4 (ATF4) in hepatoma cells [125]. Another study demonstrated that the circular RNA circ0097009 is significantly upregulated in HCC cell lines and tissues and acts as a competing endogenous RNA to regulate the expression of SLC7A11, a key regulator of cancer cell ferroptosis, by sponging miR-1261 in HCC [129]. Interestingly, expression profiles of genome-wide circRNAs in three pairs of HCC cell lines (before and after sorafenib treatment) revealed that circular RNA hsa_circ_0008367 could positively regulate sorafenib-induced ferroptosis via suppressing ALKBH5-mediated autophagy inhibition [130]. Finally, Zhang et al. found RNA-binding protein ELAVL1/HuR-dependent ferroptosis in hepatic stellate cells [102]. 

More insights on definitive regulative mechanism of non-coding RNAs on ferroptosis in HCC could support and enhance the efficiency of HCC treatment in the coming years.

## 5. Conclusions

To conclude, ferroptosis seems to harbor potential prognostic and anti-cancer properties in HCC. Despite its dualistic role in the liver, where ferroptosis might be involved in the development of liver pathologies, substances such as RSL-3 or haloperidol can induce ferroptosis, attenuate carcinogenesis and sensitize HCC cells to commonly used therapies. Furthermore, ferroptosis-associated genes show promising features in HCC prognosis and classifying HCC-patients, highlighting a future application in clinical practice. In recent years, ferroptosis has become an interesting and attractive potential approach for cancer treatment in various tumor entities. The concept of ferroptosis was suggested the first time in 2012 by Dixon, characterized by lipid peroxidation and a distinction from apoptosis and necroptosis morphologically, genetically, and mechanistically. Since then, inducing ferroptosis experimentally with FINs, such as RSL-3 or Erastin, shows promising results in targeting cancer cells and could therefore display an alternative type of cell death induction beside the well-known apoptosis. 

In cancers that display resistance towards common therapeutic strategies and are highly metastatic, GPX4 and NRF2, two factors influencing ferroptosis negatively, seem to drive cancer resistance [131]. Therefore, therapy-resistant cancers are more vulnerable to ferroptosis, highlighting a potential role of ferroptosis in drug resistance circumvention [131]. 

In HCC, the dismal outcome, caused by acquired resistance towards common therapy, elicits an urge for alternative therapeutic options. So far, current studies display promising anti-HCC activity, as well as the circumvention of sorafenib-resistance.

Although ferroptosis based therapy is promising for cancer in general, current results were only obtained in vitro [19]. That is because RSL-3 and erastin display specific solubility and metabolic properties, which do not yet allow for in vivo use [19]. Furthermore, ferroptosis induction in HCC can also damage other healthy cells and tissues due to the unspecificity of FINs [132]. Moreover, FINs can induce cell death and DNA damage in healthy bone marrow cells, and undesirable side effects in ferroptosis vulnerable organs such as the heart and the kidney are observed [19,133]. Another important point that needs to be elucidated is the dual role of ferroptosis in HCC. On the one hand, ferroptosis contributes to liver pathogenesis and cancer development and on the other hand, ferroptosis can hamper carcinogenesis, if HCC is established. 

One alternative way to circumvent potential side effects of FINs, is the usage of nanoparticles and exosomes that could directly deliver the substances, together with other cancer-therapeutics, to the tumor and induce successfully ferroptosis. Another approach might be to target miRNAs/long non-coding RNAs in HCC, as it has already been demonstrated that miRNAs (miR-214-3p), for instance, can influence ferroptosis positively.

Noncoding RNAs that are associated with ferroptosis might also serve well as potential prognostic biomarkers to predict the overall survival of HCC patients. Furthermore, current evidence suggests, aside from non-coding RNAs, ferroptosis-associated genes and scoring systems in HCC can be used as diagnostic and predictive biomarkers to identify, for instance, specific HCC subgroups to further evaluate eligibility for targeted therapy, especially for immunotherapy. Therefore, ferroptosis in HCC shows not only anti-cancer properties but may also qualify for potential translation in the clinical practice, as diagnostic and predictive biomarkers.

In this review, we aimed to give a comprehensive and up-to-date status of ferroptosis in HCC, encompassing its role in carcinogenesis and pathogenesis, its usability for treating this malignant disease and its applicability in clinics.

Therefore, further studies and intensive research regarding harmful side-effects of ferroptosis induction and the respective FINs as well as the understanding of the exact mechanisms of ferroptosis in HCC need to be accomplished to enable a future potential therapeutic and clinical application of ferroptosis in HCC.

## Figures and Tables

**Figure 1 cancers-14-01826-f001:**
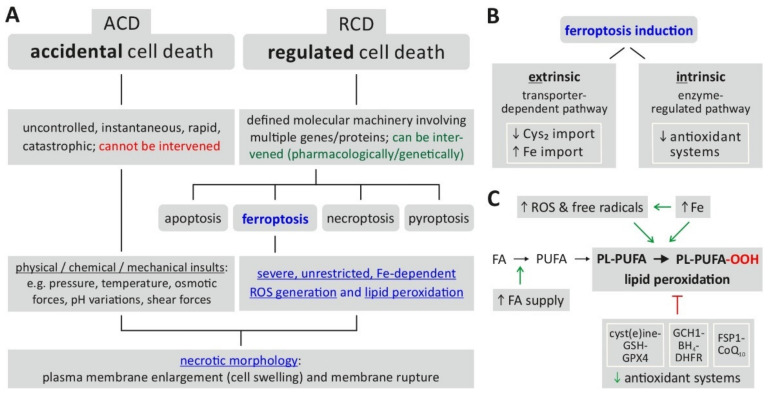
Ferroptosis—classification, characteristics, induction. (**A**) Ferroptosis is classified as a form of regulated cell death characterized by severe, uncontrolled lipid peroxidation and loss of cell membrane integrity. (**B**) The prerequisites for induction of ferroptosis include the generation of reactive oxygen species (ROS) fueled by an increase of intracellular iron (Fe), which cause oxidation of membrane lipids. The changes are enabled (facilitated) by a reduced activity of cellular antioxidant defense mechanisms. (**C**) Ferroptosis induction is further classified into an extrinsic or intrinsic pathway. Abbreviations: ACD = accidental cell death, BH4 = tetrahydrobiopterin, CoQ10 = coenzyme Q10, Cys2 = cystine, DHFR = dihydrofolate reductase, FA = fatty acid, FSP1 = ferroptosis suppressor protein 1, GCH1 = GTP cyclohydrolase 1, GPX4 = glutathione peroxidase 4, GSH = glutathione (reduced), PL-PUFA = PUFA-containing phospholipid, PL-PUFA-OOH = phospholipid hydroperoxide, PUFA = polyunsaturated fatty acid, RCD = regulated cell death, ROS = reactive oxygen species. Based on: [5,6,10,12].

**Figure 3 cancers-14-01826-f003:**
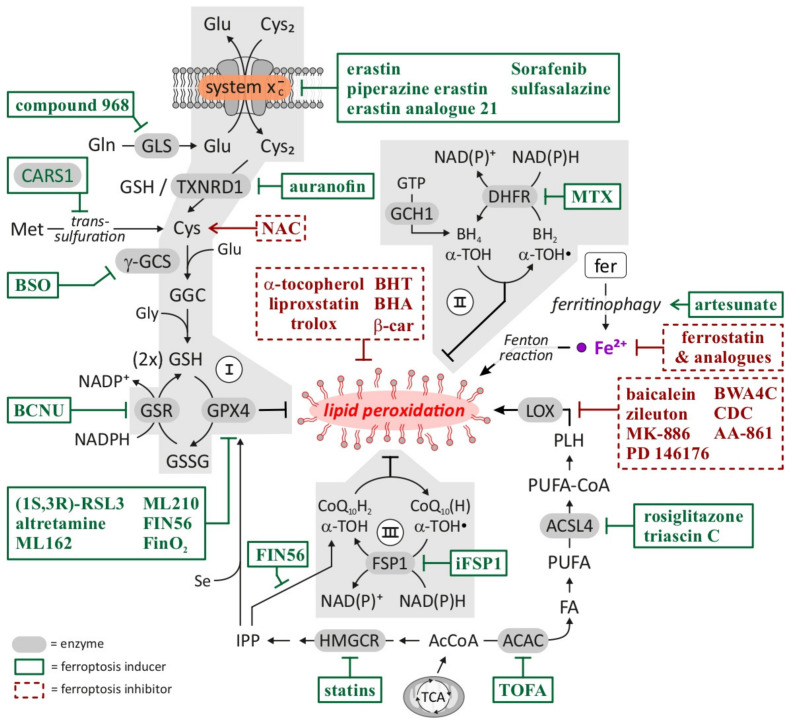
Ferroptosis—pathways inhibiting/promoting ferroptosis and pharmacological targets/drugs. Highlighted in green (boxes) are compounds that facilitate or trigger induction of ferroptosis including ‘classical’ FINs (ferroptosis inducers). Consistent with described molecular sequence or ferroptosis, compounds which either increase intracellular GSH (e.g., NAC), inhibit lipid peroxidation (e.g., α -tocopherol), reduce cellular iron (e.g., ferrostatin) or prevent synthesis of substrates for lipid peroxidation (e.g., baicalein) prevent ferroptosis induction and are highlighted in red (boxes). Abbreviations: ACAC = acetyl-CoA carboxylase, AcCoA = acetyl-coenzyme A, ACSL4 = acyl-CoA synthetase long-chain family member 4, α -TOC = α -tocopherol, α -TOC· = α -tocopheryl radical, β -car = β -carotene, BCNU = 1,3-bis-(2-chloroethyl)-1-nitrosourea, BH_2_ = dihydrobiopterin, BH_4_ = tetrahydrobiopterin, BHA = butylated hydroxyanisole, BHT = butylated hydroxytoluene, BSO = buthionine sulfoximine, CARS1 = cysteinyl- tRNA synthetase 1, CDC = cinnamyl-3,4-dihydroxya- cyanocinnamate, CoQ_10_(H) = CoQ_10_ (oxidized), ubiquinone, CoQ_10_H_2_ = CoQ_10_ (reduced), ubiquinol, Cys_2_ = cystine, DHFR = dihydrofolate reductase, FA = fatty acid, fer = ferritin, FSP1 = ferroptosis suppressor protein 1, GCH1 = GTP cyclohydrolase 1, GGC = γ -glutamylcysteine, γ -GCS = γ -glutamylcysteine synthetase, Gln = glutamine, Glu = glutamate, Gly = glycine, GPX4 = glutathione peroxidase 4, GSH = glutathione (reduced), GSR = glutathione reductase, GSSG = glutathione (oxidized), HMGCR = 3-hydroxy-3-methylglutaryl CoA reductase, iFSP1 = inhibitor of FSP1, IPP = isopentenyl pyrophosphate, LOX = lipoxygenase, Met = methionine, MTX = methotrexate, NAC = N-acetylcysteine, PLH = phospholipid, PUFA = polyunsaturated fatty acid, PUFA-CoA = PUFA-coenzyme A, Se = selenium, TCA = tricarboxylic acid, TOFA = 5-(tetradecyloxy)-2-furoic acid, TXNRD1 = thioredoxin reductase 1. Based on: [12,14,19,30,31,32].

**Table 1 cancers-14-01826-t001:** Ferroptosis inducer and their major mode of action in HCC.

Substance	Drug Name or Synonym ^a^	Mode of Action	In-Vitro/In-Vivo/In-Situ	References
Sorafenib **^b^**	Nexavar^®^	Inhibition of system xc-	✓/-/-	[78,79,80,81,82,83]
Quiescin Sulfhydryl Oxidase 1	QSOX1	Inhibition of NRF2	✓/✓/-	[84]
Artesunate	Arsumax	Ferritin degradation (in combination with sorafenib)	✓/✓/-	[85]
Eradicator of RAS and ST-expressing cells	Erastin	Inhibition of system xc-	✓/-/-	[86]
Ras-selective lethal small molecule 3	RSL-3	Inhibition of GPX4	✓/✓/-	[87]
Saponin Formosanin C	NSC 306864	Induction of ferritinophagy	✓/-/-	[88]
Heteronemin	CHEMBL514498	Reduction of GPX4 expression	✓/✓/-	[89]
Solasonine	Tomatine solaradixine	Suppression of GPX4	✓/✓/-	[90]
ZZW-115	HY-111838	Inhibition of NUPR1	✓/✓/-	[91]
Haloperidol	Haldol^®^	Inhibition of S1R	✓/✓/-	[92]

^a^: According to Pub(C)hem (see https://pubchem.ncbi.nlm.nih.gov/, last access on 30 March 2022); in-vitro: cell culture, in-vivo: tumor xenograft animal model, in-situ: human tumor specimen. ^b^: In the context of ferroptosis.

**Table 2 cancers-14-01826-t002:** Overview of in silico analysis of ferroptosis genes to develop a predictive and prognostic ferroptosis scoring system for HCC and CCC based on publicly accessible database sets.

Year	Database(s) (Dataset)	Basic Cluster Description	Predictive and/or Prognostic Aspects of the Ferroptosis Cluster	Ref
2021	TCGA: LIHC GEO: GSE76427	Ferrcluster A: “Olfactory transduction” and “cardiac music contraction”. Ferrcluster B: “mTOR signaling pathway” and “neurotrophin signaling pathway”. Ferrcluster C: “adipokine signaling pathway”, “tyrosine metabolism” and “PPAR signaling pathway”	Ferrcluster B: Overall survival ↓ High ferrscore group: Survival ↓, Programmed cell death 1 (PD-1) mRNA expression ↑, efficacy of PD-1 or PD-1 plus CTLA4 (cytotoxic T-lymphocyte associated protein 4) inhibitors ↓.	[112]
2021	TCGA ICGC	Ferroptosis-H and Ferroptosis-L: According to ferroptosis gene expression and methylation	Ferroptosis-H: Overall and disease-specific survival ↓	[113]
2021	GEO TCGA ICGC	C1: Metabolism low, immunity high subtype. C2: Metabolism high, immunity low subtype.	C1: Prognosis ↓ C1: Patients with clinical characteristics such as younger, female, advanced stage, higher grade, vascular invasion.	[114]
2020	GEO: GSE14520/GPL3921 TCGA	Low and high group: Comprehensive index of ferroptosis and immune status (CIFI).	High CIFI: Prognosis ↓	[115]
2021	TCGA	Low-risk and high-risk groups: 2 ferroptosis-related mRNAs and ferroptosis-related lncRNAs	Higher risk group: Prognosis ↓ Higher risk group: Differences of tumor microenvironment, immune cell infiltration as well as tumor-related pathways	[116]
2021 ^a^	TCGA-CHOL GEO: GSE107943 EMBL-EBI: E-MTAB-6389	Low and high group: Ferroptosis-related weighted coexpression gene network and model construction.	Higher risk group: Prognosis ↓	[117]

a…tumor entity = CCC, all other studies = HCC. Abbreviations: CCC = cholangiocarcinoma, EMBL-EBI = European Molecular Biology Laboratory—European Bioinformatics Institute, GEO = Gene Expression Omnibus, HCC = hepatocellular carcinoma, ICGC = International Cancer Genome Consortium, TCGA = The Cancer Genome Atlas. ↓ means and stands for "less" or "lower" in the relevant context.

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
