# Peer review of "Ferroptosis in Hepatocellular Carcinoma: Mechanisms, Drug Targets and Approaches to Clinical Translation"

_cancers, 2022, doi:10.3390/cancers14071826_

Round 1

Reviewer 1 Report

A timely article by Dr. Neureiter and the group elaborates the role of  Ferroptosis in Hepatocellular Carcinoma (HCC). It is a very well-documented review article in the field of cancer therapeutics and ferroptosis. 

Though few points need to be addressed before it is ready for acceptance. They are as follows:

  1. It has been discussed recently how oncogenic KRAS might play some role in ferroptosis and lipid biogenesis (PMCID: PMC8045781), while it has also been shown recently that mutant KRAS activates NRF2 antioxidant pathways which result in chemorsistance. It has also been shown NRF2 activates glutaminolysis and glutamine deprivation caused GPX4 level reduction. It is also an interesting aspect that NRF2 plays a significant role in lipid peroxidation and ferroptosis (PMCID: PMC7185043 and PMCID: PMC6859567). Oncogenes might play some role in ferroptosis. This aspect of the role of glutamine metabolism in regulating GPX4 might be one of the future areas of research in the field. It will be worthwhile to add a few lines on this aspect.  
  2. Authors should add a table mentioning the drug names which are in the clinical trial stage targeting ferroptosis.
  3. It will be worthwhile if a few lines would be added to the historical timeline of ferroptosis discovery.
  4. It has been shown (PMCID: PMC8073197) stress granules might play some role in ferroptosis while both of them are involved in regulating chemoresistance. There might be some possibilities that the interconnection between ferroptosis and stress granules might have some implication in radiotherapy resistance. Authors should add a few lines on this as one of the future aspects of the field.
  5. It has been shown recently that GSTZ1 enhanced sorafenib-induced ferroptosis by inhibiting the NRF2/GPX4 axis in HCC cells (PMID: 33931597). Authors should discuss this from their perspective.

Reviewer 2 Report

The review article “Ferroptosis in Hepatocellular Carcinoma: Mechanisms, Drug Targets and Approaches to Clinical Translation” by Bekric et al., provides a comprehensive insight on the mechanism of ferroptosis in HCC and the mechanistic role of several ferroptotic agonists in HCC. The manuscript shows that the team has great insight in the field of ferroptosis which allowed them to write this insightful article. The review is well organized and written and displays a broad readership to get a summary of ferroptosis in HCC. I have few comments and suggestions.

  1. Authors should elaborate the introduction part and include the recent references related to this field example PMID 30925886, 34970553, 34611144 and 32015325.
  2. Expand most of the acronyms in their first appearance in the text since most of them are abbreviated in figure 2 legend (line 133-144). It’s better to include them in the main text whenever they appear first.
  3. Line 239 “Iron an Ferroptosis in Hepatocytes and Liver Pathologies” change it to “Potential role of ferroptosis in Hepatocytes and Liver Pathologies” for better readership.
  4. Line 262 change “up to” to “upto” and mention the iron content as “iron/g dry weight of liver” and it would be more appropriate to include few lines about iron absorption and toxicity in liver.
  5. Line 265 whether is this Ferritin or Apoferritin, make sure. As for as my knowledge it should be Ferritin. You can quote or refer to PMID 31949017.
  6. Line 293 Deferiprone and Trolox are not used in the quoted reference PMID 20427778 where the authors used Pioglitazone and Vitamin E for their study. Include appropriate reference here for Deferiprone and Trolox.
  7. Line 327 why the reference 35 is included here for the table since all the references quoted in the table are provided in the table 1 itself.
  8. Line 360-361 add appropriate references.
  9. Line 439 change “sigma receptor 1” to “sigma-1 receptor”
  10. Line 438 authors should discuss the role of GSTZ1 on NRF2/GPX4 axis in the context of HCC by including these references PMID 33931597, 31267557 and 31666108.
  11. Line 629 comprehensively conclude the review remove the lines 641-642. And remove “for example” in line 637 instead start with “Cancers that display……….”
  12. Minor typo errors were found for example “et Al.” on lines 282, 304, 306. Line 242 change “830.000” to “830,000”. Change line 252 “those” to “these”. Change lines 347 “acts” to “are the”. Line 350 and to “thereby induces”. Line 415 change “potential” to “potentiality”. Line 670 “uptodate” to “up-to-date”.

Round 2

Reviewer 1 Report

All concerns have been addressed, ready for acceptance.